# Opioid Addiction, Genetic Susceptibility, and Medical Treatments: A Review

**DOI:** 10.3390/ijms20174294

**Published:** 2019-09-02

**Authors:** Shao-Cheng Wang, Yuan-Chuan Chen, Chun-Hung Lee, Ching-Ming Cheng

**Affiliations:** 1Jianan Psychiatric Center, Ministry of Health and Welfare, Tainan 717, Taiwan; 2Department of Mental Health, Johns Hopkins Bloomberg School of Public Health, Baltimore, MD 21205, USA; 3Program in Comparative Biochemistry, University of California, Berkeley, CA 94720, USA; 4Department of Informative Engineering, I-Shou University, Kaohsiung 840, Taiwan; 5Department of Food Nutrition, Chung Hwa University of Medical Technology, Tainan 717, Taiwan; 6Department of Natural Biotechnology, NanHua University, Chiayi 622, Taiwan

**Keywords:** opioid addiction, opioid dependence, opioid receptors, methadone, buprenorphine, naloxone, GWAS, polygenic risk score, CRISPR, Medical treatment

## Abstract

Opioid addiction is a chronic and complex disease characterized by relapse and remission. In the past decade, the opioid epidemic or opioid crisis in the United States has raised public awareness. Methadone, buprenorphine, and naloxone have proven their effectiveness in treating addicted individuals, and each of them has different effects on different opioid receptors. Classic and molecular genetic research has provided valuable information and revealed the possible mechanism of individual differences in vulnerability for opioid addiction. The polygenic risk score based on the results of a genome-wide association study (GWAS) may be a promising tool to evaluate the association between phenotypes and genetic markers across the entire genome. A novel gene editing approach, clustered, regularly-interspaced short palindromic repeats (CRISPR), has been widely used in basic research and potentially applied to human therapeutics such as mental illness; many applications against addiction based on CRISPR are currently under research, and some are successful in animal studies. In this article, we summarized the biological mechanisms of opioid addiction and medical treatments, and we reviewed articles about the genetics of opioid addiction, the promising approach to predict the risk of opioid addiction, and a novel gene editing approach. Further research on medical treatments based on individual vulnerability is needed.

## 1. Introduction

Opioid addiction is a chronic mental illness that causes the addicted individuals to experience many relapses and remissions throughout their life, and they suffer from many uncomfortable symptoms, including tolerance development and withdrawal [1,2]. Compared to smoking and alcohol consumption, opioid addiction is less common; however, it has imposed a heavy burden on both healthcare systems and the criminal justice system [3,4]. The opioid crisis costs the United States billions of dollars annually, particularly due to the recent prescription opioid epidemic. The treatment of opioid addiction with medications can play an important role in preventing relapse and enabling the addicted individuals to be stable enough to work; thus, they can have longer periods of abstinence [5,6].

Over the last two decades, the opioid epidemic or opioid crisis in the United States has raised public awareness, and effective interventions are urgently needed [7]. Medications for opioid addiction such as methadone and buprenorphine are used to treat addicted individuals by reducing the intensity of withdrawal and craving symptoms, and naloxone is used to treat opioid overdose or opioid intoxication [8,9,10]. Though effective medications for opioid addiction are available, relapse and remission are still common among addicted individuals. The risk of relapse is heightened due to the craving feeling with terrible withdrawal symptoms, as well as neurobiological changes in brain caused by the repeated abuse of opioids.

Several studies have suggested that the heterogeneity of the population may be associated with the development of addiction and its longitudinal outcomes, such as relapse or remission. Alcoholics have lower frequencies of the alcohol dehydrogenase and aldehyde dehydrogenase alleles than non-alcoholics, suggesting that these genetic variations influence the risk of developing alcoholism [11]. Hill et al. suggested that patterns of alcohol use can be classified into four trajectory groups: “Early highs,” “increasers,” “late onsetters,” and “non-bingers” [12]. Jackson et al. found four trajectory classes, using growth mixture modeling with five indices of alcohol consumption [13]. Genberg et al., using latent growth curve modeling and the AIDS Linked to the IntraVenous Experience longitudinal cohort, found that I.V. drug users can be categorized into five different groups with unique trajectories: Persistent injection, frequent relapse, early cessation, delayed cessation, and late cessation [14].

In this article, we summarize the biological mechanisms of opioid addiction and opioid receptors. We also review previous articles about the medical treatments for opioid addiction and the genetic variants of addiction susceptibility, trying to link the novel gene-editing therapy to the opioid addiction.

## 2. Opioid Addiction, Biological Mechanism and Medications

Most individuals begin to use opioids for euphoric feelings or pain relief; then, tolerance develops due to the desensitization of opioid receptors, leading to an uncontrolled intake. The symptoms of opioid tolerance and withdrawal, with uncontrolled intake and craving, are the core symptoms of opioid addiction. The withdrawal symptoms include severe muscle ache, bone pain, tearing, runny nose, yawning, diarrhea, abdominal cramps, agitation, anxiety, and sweating. Many addicted individuals used opioids again to alleviate these intolerable feelings [1,15,16,17]. Volkow et al. reviewed articles related to opioid addiction and found that the molecular mechanisms of opioid-induced tolerance and physical dependence play an important role in opioid addiction [18]. After repeated opioid administration, the opioid receptor regulation such as desensitization and internalization, started, as did opioid receptor tolerance, involving several cellular processes such as the upregulation of cyclic AMP/protein kinase A and cAMP response element-binding signaling [19]. In addition, cravings for drugs, diminishing self-control, and strong responses to drug-associated stimuli are associated with the cellular and molecular change of the glutamatergic projection in the prefrontal cortex and the basal ganglia region, such as alterations in G protein signaling in the prefrontal cortex and increased presynaptic glutamate release in the accumbens [20,21].

People with substance abuse disorders experience changes in behavior that impact their ability to function, and these changes may continue even after someone has quit using. Even worse, long-term addiction impairs intellectual function. Alcoholics, for example, are at a high risk of Wernicke–Korsakoff Syndrome [22,23]. Intellectual functional impairment—specifically, cognitive impairment and poor judgement—can be found in addicts who regularly use opioid and amphetamines [24]. This functional impairment often translates into their inability to hold down a steady job or to maintain healthy relationships; thus, effective interventions for addiction are needed to stop and/or prevent cognitive deterioration.

Currently, there are three groups of medications for opioid addiction, including opioid receptor full agonists, partial agonists, and antagonists [25]. Methadone is a full opioid receptor agonist; patients who received methadone maintenance treatment (MMT) have less intensity of withdrawal symptoms such as muscle ache and bone pain; thus, they are less likely to use opioids again to prevent the terrible feeling [26]. Buprenorphine, a partial opioid receptor agonist, was developed as an alternative to methadone; it does not stimulate the μ opioid receptor (MOR) to the same degree as methadone, so the addicted individuals under buprenorphine treatment are less likely to have respiratory depression and euphoria [27]. Naloxone and naltrexone are antagonists, targeting all subtypes of the opioid receptor, binding to opioid receptors and blocking the agonist; thus, they do not cause similar effects to agonists such as respiratory depression and euphoria. Naloxone’s affinity is the highest for the MOR and is used to treat respiratory depression in addicted individuals with a severe opioid overdose. Naltrexone binds to both the MOR and the κ opioid receptor (KOR), and a long-acting form of injectable naltrexone has a continuous effect to decrease the frequency and dosage of heroin use [28].

### Opioid Receptors

Opioid receptors are G protein-coupled receptors distributed across the brain, spinal cord, skin, and gastrointestinal tract [29,30,31,32]. μ, κ, and δ are subtypes of the opioid receptor, sharing a common analgesic effect in brain, and each of them has their unique effects such as euphoria and respiratory depression for the MOR, dysphoria for the KOR, and anxiolysis for the δ opioid receptor (DOR) [33,34].

The MOR was the first discovered opioid receptor and can trigger euphoria; therefore, it is essential for brain reward circuits which are highly dynamic [35,36,37], and it also plays an important role in goal-directed behavior such as drug-seeking behavior for pleasure. As opioid addiction becomes severe, cognition impairment causes addicted individuals to make poor decisions, shifting goal-directed behaviors to habitual behaviors such as repetitive drug seeking and/or craving [38,39]. A common polymorphism (A118G) in the MOR gene (OPRM1) is associated with social rejection and social hedonism in the study of human neural sensitivity [40], and recent mice studies suggest that the MOR is also associated with social attachment and anhedonia [41]. As opioid tolerance develops, compulsive seeking for more opiates and uncontrolled intake emerges; as a result, the homeostasis of brain reward circuits and the ability to maintain fair social functioning becomes compromised [42]. Recent studies have demonstrated that the effect of the MOR varies with age; the adolescent rodents have more positive reinforcement of the MOR and less opioid withdrawal symptoms, compared with the adult rodents, and this finding is consistent with the notion that addictive behaviors are initiated more among teenagers [43,44]. 

The KOR triggers anti-reward effects and produces dysphoria [45]. Prolonged exposure to drugs of abuse can change the brain reward circuits and enhance the KOR function through the hypothalamic–pituitary–adrenal axis, promoting opioid addiction relapse [46,47]; furthermore, stress due to prolonged exposure to drugs can also produce a dysphoric feeling. In brief, the intensified stress due to long term exposure to opioids can enhance the function of the KOR, producing a dysphoric mood and leading to relapse. The DOR binds with enkephalins, and it can then reduce levels of anxiety and attenuate depressive symptoms [48]. 

## 3. Medications for Opioid Addiction

Among addicted individuals, one reason for the continuous use of opioids is to stave off withdrawal symptoms such as muscle ache and bone pain for few hours after using, because heroin and morphine have short half-lives [49,50]. Opioid agonist therapy can stave off withdrawal symptoms, so the addicted individuals under treatment are able to work and have a stable life. 

### 3.1. Methadone, Buprenorphine, and Naloxone

Methadone is a full MOR agonist, has some agonist effects on the KOR, and is also a possible DOR agonist [51]. Methadone causes fewer withdrawal symptoms and is less reinforcing due to its longer half-life [52]; consequently, the addicted individuals under the MMT are no longer preoccupied with compulsive drug seeking and craving, and they are more willing to remain in treatment, improving their psychosocial function with help from the health care providers [53,54,55]. Addicted individuals receiving MMT and consuming alcohol and/or sedatives at the same time are at high risk of respiratory depression [56]. 

Buprenorphine is a partial agonist with a high affinity for the MOR, a partial KOR agonist or functional antagonist, and a weak DOR antagonist [57,58]. Compared to the full MOR agonist, buprenorphine has a ceiling effect on the MOR, providing less euphoric feelings and having less risk of respiratory depression; thus, it is safer than methadone and adequate to alleviate opioid withdrawal symptoms [59]. Buprenorphine, in combination with naloxone is used commonly to treat opioid addiction in the United States, and the addicted individuals are less likely to inject it due to severe opioid withdrawal [60]. 

Naloxone and naltrexone are both opioid antagonists. Naloxone is a non-selective and competitive opioid receptor antagonist, and it is used for acute opioid intoxication, reversing the most dangerous side effects such as respiratory depression [61]. Long-acting injectable naltrexone can block opioid receptors, alleviating craving and decreasing the risk of opioid overdose [62,63]. 

### 3.2. Pharmacogenomics for Opioid Addiction

As we mentioned above, methadone, buprenorphine, and naloxone have been proven to be effective for opioid addiction, but opioid addiction relapse and recurrent are still very common. Personalized medicine and pharmacogenomics might be promising effective treatments for opioid addiction. The pharmacogenetics of opioid analgesics and pain treatment have raised the attention of researchers in recent years [64,65]; however, these articles have focused on the mechanisms of opioid receptors and analgesic effects instead of opioid addiction. In the next paragraph, we review the articles related to genetic susceptibility on addiction and summarize the genes related to opioid addiction, providing possible goals of pharmacogenomics for opioid addiction.

## 4. Genetic Susceptibility on Addiction

The genetic underpinnings of substance use disorders are frequently studied. Classic genetic approaches such as twin, family and adoption studies show that there are significant genetic influences on drug addiction. Heritability is the proportion of observed differences on a phenotypic trait among individuals of a population that are due to genetic differences, and the heritability of addiction has been estimated at 0.4–0.6 [66,67]. In recent years, huge advances in computer technology have made molecular genetic approaches possible. Linkage studies, genetic association studies, and genome-wide association studies can identify and locate associated genes. There have now been a number of studies on addiction behaviors using molecular approaches [68,69,70,71,72]. Classical genetic approaches have shown that addiction is heritable; molecular genetic approaches suggest that specific addiction-related behaviors are associated with specific genes. 

### 4.1. Classic Genetic Research and Molecular Genetic Research on Addiction

Genetic epidemiology is a rapidly expanding field of research [73]. Both classical genetic research and molecular genetic research provide valuable information related to substance use disorders. This section provides a review of the latest genetic research on disorders involving tobacco, alcohol, and opiates. The subsequent section discusses the possible genetic links between opioid addiction and specific behavior patterns. 

Kreek et al.’s review of the research on genes and addiction offered a three-domain model that included genetics, diverse environmental factors, and drug-induced effects [74]. In 1960, Kaij et al. conducted the first study of alcoholism in twins, and in 1966, Partanen et al. conducted a similar twin study which further explored the associations between intelligence, personality, and alcohol consumption. These were the earliest studies proposing that specific addictions were heritable or influenced by genes [75,76]. An adoption study by Cloninger et al. concluded that genes influence alcohol abuse, since adopted-away probands had a greater resemblance to their biological relatives than their adoptive families [77]. Furthermore, Cloninger et al.’s adoption study provided a classic approach to disentangle the influence of genetics from that of environmental factors [77]. In 1988, Merikangas et al. reported an eight-fold increase in the odds of drug disorders among the relatives of probands with drug disorders, with the greatest odds ratio observed for addiction to the same substance [78]. Tsuang’s twin study posited that both the environment and genes influence a person’s susceptibility to drug abuse; Tsuang also found that all commonly abused drugs—opiates, marijuana, sedatives, psychedelics, and stimulants—had an overall genetic variance of 0.3–0.5. Heroin had the greatest overall genetic variance, 0.54, and a shared genetic variance, 0.2, with other drugs. However, most of these drugs only had a low variance for specific genetic factors, and only heroin had the greatest specific genetic variance at 0.4, indicating there could be unique genetic factors affecting opioid abuse [67]. Kendler and colleagues, in their seminal twin study on substance use disorders, found that lifetime drug use of cannabis, cocaine, hallucinogens, sedatives, stimulants, and opiates had a range of additive genetic variance, or heritability, of 0.3–0.5 [66]. From twin studies to adoption studies, from alcohol to other substances, these classic genetic studies provide solid evidence that genetics plays an important role in substance use disorders. Subsequently, molecular genetic studies have explored heritability on a deeper level. In the past twenty years, with huge advances in computer technology and genomic array technology, molecular genetic approaches have identified or located specific associated genes.

### 4.2. The Highest Heritability: Opioid Addiction 

Tsuang et al.’s twin study found that heroin had the greatest overall genetic variance, 0.54, and a shared genetic variance, 0.2, with all drugs [67]. Wilen and colleagues’ study of families in which parents were opioid or alcohol-dependent reported that children of addicts experienced higher rates of psychopathology including mood, anxiety, and substance use disorders [79]. An adoption study by Cadoret et al. found that heritable biological and environmental factors correlated with substance use. These results were consistent with twin studies [80]. 

Scientists have started using molecular approaches in genetic epidemiology to identify the location of addiction-associated genes. Linkage studies are family-based studies that link to specific regions of the genome rather than a particular gene; they target phenotypes such as physical traits or specific diseases [81]. Until twenty years ago, linkage studies could detect only a limited number of candidate genes because the technology was not sufficiently advanced; now, with much more powerful computers, genome-wide linkage studies are possible. Gelernter et al. conducted a genome-wide linkage scan for opioid dependence and found that chromosomes 2 and 17 are associated with opioid dependence [82]. Lachman and colleagues conducted another genome-wide linkage study of opioid dependence and found that a specific region of chromosome 14q is associated with opioid dependence [83]. Genome-wide association studies have found, in a hypothesis-free approach, that single-nucleotide polymorphisms (SNPs) throughout the genome are associated with a specific phenotype by comparing the affected individuals to non-affected individuals [84,85]. Wetherill et al. conducted a genome-wide association study examining the association between candidate genes and substance dependence and found that one SNP, rs2952621 in the uncharacterized gene LOC151121 on chromosome 2, and another SNP, rs2567261 in ARHGAP28 (Rho GTPase-activating protein 28), are associated with substance dependence [86]. In a separate genome-wide associations study, Gelernter and colleagues found that SNPs from multiple loci-KCNG2*rs62103177 which involved potassium signaling pathways were associated with opioid dependence [87].

### 4.3. Genetic Susceptibility and Psychological Traits

Some behavior patterns such as impulsivity, risk taking and stress response, which are due to specific personality and physiological traits, may make some people more prone to addictive disorders. These patterns may be partially influenced by genetic variation. Moreover, differences in personality and physiological traits may affect different stages of addiction. These stages of addiction are chronologically defined as the initiation of drug use, regular drug use, abuse/dependence, and relapsed use [88]. Clearly, many genes have been found to be associated with addiction. The focus of this work is on genes associated with heroin addiction. These genes can be classified into two gene systems: The dopaminergic system and the MOR system [74]. Several single-nucleotide polymorphisms in the dopaminergic system are associated with heroin addiction. They include rs4680, rs1800497, rs1800955, rs1611115, rs1079597, rs747302, rs1800498, and rs936462 [89,90,91]. The dopamine D4 receptor gene was also found to be associated with novelty-seeking, which is further associated with risk-taking [92,93]. Candidate genes OPRM1, rs1799971, rs7997012, and rs540825 in the MOR system are associated with opioid dependence [94,95,96]. We summarized the above findings and listed the characteristics of genes related to heroin/opiate dependence, including protein product, system, location on chromosome, and associated SNPs in Table 1.

Taken together, given the evidence for the highest heritability in opioid dependence plus lone successes in alcohol and tobacco for genetic variations among users, many researchers have been trying to find genetic variations among individuals with opioid addiction. As we previously described, many genes have been linked to opioid addiction; thus, opioid addiction is more likely a complex disease rather than a single gene disorder. Polygenic risk score analysis may be a promising tool to evaluate the association between behavioral phenotypes and genetic markers across the entire genome and to predict the risk of opioid addiction. 

### 4.4. Polygenic Risk Score for Opioid Addiction

The polygenic risk score analysis creates a score from the top SNPs from a genome-wide association study (GWAS) on a target phenotype, and it is commonly used to examine common disease-common variant hypotheses. 

The polygenic risk score analysis is a statistical approach which is used to summarize genetic effects among a group of SNPs which do not have significant associations with diseases/traits. The approach is based on the assumption that although many SNPs do not reach significance after correcting for genome-wide testing, the tail of the distribution of *p*-values less than some target threshold will be enriched for a true signal. A GWAS is conducted first on a training sample, and the *p*-values of SNPs are obtained. A polygenic risk score is then constructed in an independent sample as a weighted sum scores trait-associated alleles for each subject, for different subsets of top ranking markers [97]. The first successful polygenic score analysis in/within a GWAS was applied to schizophrenia and bipolar disorder [98]. This approach has two possible applications. First, the polygenic scores can be used to determine the association between a disease/trait and selected SNPs. Second, the polygenic scores can be used to predict individual disease/trait values. This approach has attracted much attention and has been used with several common and complex disease-including multiple sclerosis, cardiovascular risk, and rheumatoid arthritis [99,100,101]. 

## 5. CRISPR Gene Editing for Opioid Addition 

Recently, clustered, regularly-interspaced short palindromic repeats (CRISPR) has been extensively used in basic research and potentially applied to the treatment of human diseases such as virus latent infections, genetic diseases, neurodegenerative disorders, cancers, and mental illness [102,103]. CRISPRs are loci that contain multiple, short, direct repeats of DNA sequences in bacteria. Each repeat consists of a series of DNA sequences followed by 25–30 base pairs of DNA segments known as “spacers.” The spacers are derived from a bacteriophage or plasmid and serve as a ‘memory’ of past exposures. When the host encounters the same specific DNA sequence again, it can recognize the foreign DNA by complementarily base pairing with the stored short spacer in CRISPR RNA (crRNA). A crRNA/transactivating crRNA (tracrRNA) hybrid acts as a guide RNA for the CRISPR-associated (Cas) protein, which is an endonuclease working to cleave the invading DNA. In the CRISPR/Cas9 system, CRISPR is used to build RNA-guided genes drives to target a specific DNA sequence. By a specifically designed single guide RNA (sgRNA) and the Cas protein, the organism’s genome can be cut at most locations with the only limitation that the protospacer adjacent motif (PAM) sequence (NGG) is available in the targeting site [102,103] (Figure 1).

### 5.1. CRISPR/Cas9 as a Tool for Studying Mental Illness 

The CRISPR/Cas9 system has been successfully applied to basic research for mental illness, including opioid addiction, by modifying, activating, or suppressing specific genes of interest, mainly in the generation of transgenic rats. Rats have historically been considered as strong animal model for behavior studies; however, compared with transgenic mice, their application is limited due to the lag in the development of rat genome editing. At present, CRISPR techniques have revolutionized the genome editing of rats, and, thus, gene targeting is no longer restricted to mice [104]. Back et al. used the adeno-associated virus (AAV)-CRISPR combination system to introduce mutation in the dopaminergic neuron gene, thereby testing the function of genes in substance/drug abuse or Parkinson’s disease model. This is the first knockin rat model produced by targeting germline genes in spermatogonial stem cells [105]. 

D2 dopamine receptors (Drd2), acting as a primary target for medication, are involved in brain disorders such as drug addiction, schizophrenia and Parkinson’s disease. Yu et al. utilized CRISPR techniques to generate two knockin rat lines (Drd2::Cre and Rosa26::loxp-stop-loxp-tdTomato). By crossing these two rat lines, they produced Drd2 reporter rats that expressed the fluorescence protein tdTomato under the control of the endogenous Drd2 promoter. The generated Drd2::Cre rats could provide an excellent animal model to study the function of neuronal populations expressing Drd2 [106]. 

Ghrelin, a stomach-derived hormone, takes effects by binding to the growth hormone secretagogue receptor (GHSR) and influences various animal behaviors including drug addiction, reward, stress, and feeding. Zallar et al. established, verified and characterized a GHSR knockout (KO) Wistar rat model using CRISPR techniques. This GHSR KO Wistar rat model may represent a novel and preferred tool for studying the role of the ghrelin system in neuropsychiatric disorders and obesity [107].

### 5.2. Potential Treatment for Opioid Addiction Based on CRISPR

There have been many applications against opioid addiction based on CRISPR under research, and some of them have been successful in animal studies. The outcome in genome editing using CRISPR/Cas9 can expedite the related studies of brain disorders and potentially offers a therapeutic strategy for drug addiction by transplanting stem cells or directly modifying related genes.

Cocaine addiction, a brain disease associated with compulsive drug-seeking, is easy to relapse once exposed to the drug, even after long periods of abstention. Li et al. reported that skin epidermal stem cells can be engineered using CRISPR and then transplanted back into donor mice. The transplantation of skin cells was shown to express an active butyrylcholinesterase to hydrolyze cocaine and maintain this enzyme release for a long-term period to protect mice from the behavors of cocaine-seeking and overdose. The result revealed that the cutaneous gene therapy through skin transplants that facilitate drug elimination may provide an alternative therapeutic choice for drug addiction [108].

Gamma aminobutyric acid (GABA) type-A (GABA-A) receptors encompassing the α2 subunit (Gabra2) which express in most regions of brain play an important role in modulating inhibitory synaptic function. The variation in the Gabra2 genetic locus may cause psychiatric disorders, epilepsy, alcoholism and drug abuse. Mulligan et al. repaired the deletion of the Gabra2 gene in C57BL/6J (B6J) mice using CRISPR and completely restored the levels of Gabra2 mRNA and proteins in the mice brain. The result had crucial implications for researchers using B6J mice as their background strain to study the molecular genetics of neurobiology and behavior, and it further hinted at its possibility to treat drug abuse through Gabra2 gene therapy [109].

Though the medical treatment for opioid addiction based on CRISPR is promising for clinical applications, there are associated challenges regarding technique limitations, safety concerns and ethics issues that need to be deliberately evaluated (Table 2). Fortunately, many strategies have been developed to improve consequences and overcome most challenges [110,111,112,113,114,115,116].

## 6. Conclusions

In the United States, billions of dollars have been lost due to opioid addiction, and thousands of deaths have been caused by opioid addiction annually [117,118]. In particular, the high prevalence of opioid addiction among individuals in their late adolescence and early adulthood has imposed a heavy burden on societal structures, such as those of health care and criminal justice [119]. 

Animal studies have supported the idea that adolescent rodents are more likely to develop opioid addiction because they have more positive feedback from the MOR and less opioid withdrawal symptoms. As adolescents develop opioid addiction, drug craving and seeking can ruin individuals’ education and employment opportunities and can even lead to criminal activity resulting in incarceration; consequently, the irreversible brain function impairment causes a vicious circle where addicts initially want to get rid of drugs but then use again because they are incapable of coping with the feelings of frustration, repairing relationships with the people around them, and getting employed.

Opioid agonist and antagonist medications are proven to be effective treatments for opioid addiction. Opioid agonist therapies such as methadone and buprenorphine can reduce the intensity of euphoria and withdrawal, and opioid antagonist therapy can prevent the misuse of opioid replacement medications. As mentioned above, opioid receptors are associated with the brain reward system, and medications with a greater effect on the MOR, which is associated with rewarding stimuli, are more likely to keep addicted individuals in treatment. The aim of opioid dependence treatment is to stabilize the patients’ medical, psychiatric, legal, family, housing and employment problems. However, recurrent and relapse are common under current medical and psychological treatments. Thus, genetic research may provide an approach to reveal the mechanisms of opioid addiction and to understand individual differences in vulnerability for opioid addiction. 

Classic genetic research and molecular genetic research have both provided evidence that opioid addiction is not only environmental but also at least partly heritable; the findings of previous molecular genetic studies such as GWAS support the idea that opioid addiction is associated with specific genes which are involved with opioid receptors, the dopaminergic system, and the serotonergic system. As previously mentioned, opioid addiction is associated with specific behavior patterns such as tolerance, craving and withdrawal symptoms, and it is also associated with brain reward circuits through the opioid receptor system. The above findings suggest that opioid addiction is a complex disease. The polygenic risk scores analysis has been applied to psychiatric disorders such as schizophrenia and bipolar disorder, and the findings have supported the idea that schizophrenia and bipolar disorder are complex diseases [99,120]. Further research on applying a polygenic risk score analysis to opioid addiction and other substance use disorders is needed. 

The novel gene editing method, CRISPR, has the potential to work as an efficient tool to generate transgenic rats that have experimental advantages over mice for studying opioid addiction. For medical treatment, many therapeutic strategies including cell therapy and gene therapy are being developed for opioid addiction. If opioid addiction is an inheritable disease with genetic defects in Drd2, GHSR and Gabra2, it is reasonable to apply gene editing technologies (e.g., CRISPR) to medically and ethically prevent opioid addiction. If patients are already suffering from opioid addiction, the therapeutic outcome of using CRISPR gene editing is limited unless the patients have defective genes associated with opioid addiction. Novel CRISPR/Cas9 gene editing has led to an increased interest in treating neurodegenerative diseases such as Alzheimer’s disease and Huntington’s disease [121]. Thus, gene therapies might be effective in the treatment of opioid addiction, though most of them are currently under research and more trials/review are needed due to safety concerns and ethics issues. Regarding these cases, cell therapy based on CRISPR is recommended and potentially useful for the treatment of patients already suffering from opioid addiction. The development and implications of CRISPR gene editing technology should be investigated in as broad a context as possible. Additionally, the future research direction and application of these new findings should also be highlighted.

## Figures and Tables

**Figure 1 ijms-20-04294-f001:**
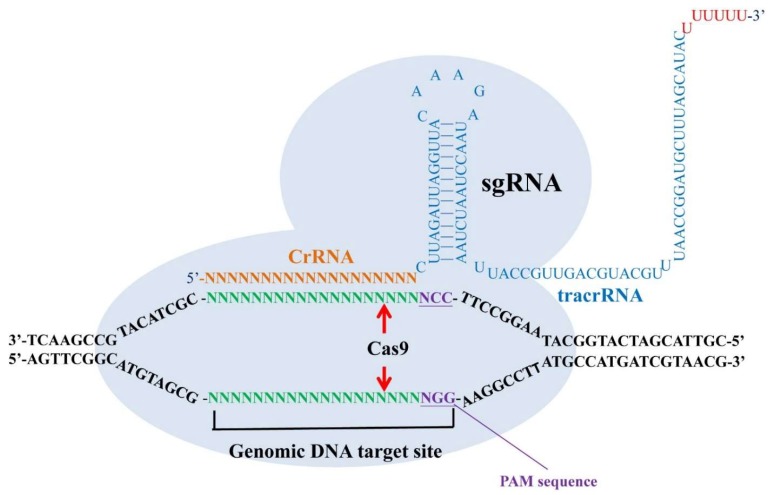
A single guide RNA (sgRNA) which is composed of clustered, regularly-interspaced short palindromic repeats (CRISPR) RNA (crRNA) and transactivating crRNA (tracrRNA) can recognize their target DNA sequence and allow the Cas9 protein to cleave 3 bp upstream of the protospacer adjacent motif (PAM) sequence (NGG) in both sense strand and antisense strand. The specific binding of sgRNAs and the cleavage of Cas9 endonuclease make a double-stranded break (DSB) with blunt ends in their genomic target sites.

**Table 1 ijms-20-04294-t001:** Characteristics of genes related to heroin/opiate dependence and addiction.

Gene	Protein	System/Function	Chromosomal Location	Associated SNP
*OPRM1*	µ opioid receptor	Opioid	6q24-25	rs1799971
*OPRK1*	κ opioid receptor	Opioid	8q11.2	rs963549 rs1051660
*DRD4*	Dopamine receptor D4	Dopaminergic	11q15.5	rs1800955, rs747302, rs936462
*TPH2*	Tryptophan hydroxylase 2	Serotonergic	12q.21.1	rs4290270 rs7963720
*HTR1B*	Serotonin receptor 1B	Serotonergic	6q13	rs130058 rs11568817
*SLC6A4*	Serotonin transporter	Serotonergic	17q11.1-q12	
*COMT*	Catechol-*O*-methyl transferase	Catecholaminergic	22q11.2	rs4680
*CYP2D6*	Cytochrome CYP450	Drug metabolism	22q13.1	

**Table 2 ijms-20-04294-t002:** Challenges of medical treatment for opioid addiction.

Challenge	Issue
Technique	Nonspecific target and cleavage (off target) Localization of function with the availability of NGG sequence Difficulty in the selection of delivery tools
Safety	Gene mutation caused by off target effects Tumorigenesis induced by *p53* gene suppression Possible insertion of foreign genes into self-body
Ethics	Complexity of risk assessment Imbalance between public interests and private interests Commercialization of therapeutics in humans Random manipulation of germline genes

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
