# Peer review of "Opioid Addiction, Genetic Susceptibility, and Medical Treatments: A Review"

_ijms, 2019, doi:10.3390/ijms20174294_

Round 1

Reviewer 1 Report

I appreciate the authors' efforts in conducting this review on a topic that people care about now and hasn't been comprehensively reviewed yet. The manuscript is well written and organized properly. My suggestions are as follows:

The authors described that polygenic risk score is potentially a method to identify the relationship between opioid addiction and other behavioral traits. I would suggest the authors to add a paragraph to review and summarize existing work applying polygenic risk score or other polygenic methods (e.g., GTCA analysis or LD score regression) to investigating opioid addiction or other substance use disorders. I would suggest the authours to add a section to briefly review current knowledge on the pharmacognomics of the treatment for opioid addiction.

Author Response

To Dear Reviewer1,

We appreciate your suggestions and added the section 4.4 Polygenic risk score for opioid addiction to discuss the existing work applying polygenic risk score to investigating opioid addiction, and also added section 3.2 Pharmacogenomics for opioid addiction to briefly review current knowledge on the pharmacogenomics of the treatment for opioid addiction.

Again, we appreciate your review to make this manuscript more complete and more comprehensive.

Reviewer 2 Report

The review article by Wang and co-workers deals with an update based on individual differences in vulnerability for opioid addiction. An optimal medical treatment following a guideline personalized would need further research. The paper is a good piece of work, it is well written and the conclusions are adequate, however the only question which should be addressed is in Table 1, where the references have not been included.  

Author Response

To Dear Reviewer 2,

We appreciate your suggestions and had rewritten the following paragraph, “We summarized the above findings and listed the characteristics of genes related to heroin/opiate dependence, including protein product, system, location on chromosome, and associated SNP in Table 1.” We listed the reference 90-97 for Table 1 and noted it in this paragraph.

Again, we appreciate your review to make this manuscript more complete and more comprehensive.

Reviewer 3 Report

ijms-569450-peer-review

Wang et al. supposed to discuss the biological mechanisms of opioid addiction, medical treatments, and reviewed articles about the genetics of opioid addiction. They also mention CRISPR gene editing for opioid addiction. Their description of opioid addiction is less detailed in both molecular and neurological backgrounds. Behavioral changes in opioid addiction accompany with structural and functional changes in neuronal connectivity. They did not mention brain circuits related to opioid addiction. Therefore, it is a little difficult to understand why gene editing treatment can be useful in opioid addition.

              Past review article discussed that “ The molecular mechanisms underlying addiction are distinct from those responsible for tolerance and physical dependence. Cardinal features of addiction include a pronounced craving for the drug, obsessive thinking about the drug, erosion of inhibitory control over efforts to refrain from drug use, and compulsive drug taking. These behavioral changes in turn are associated with structural and functional changes in the reward, inhibitory, and emotional circuits of the brain.”  Opioid Abuse in Chronic Pain — Misconceptions and Mitigation Strategies. N Engl J Med 2016; 374:1253-1263

Questions;

Desensitization, tolerance and addiction are different concepts. This review did not describe clearly biological and molecular mechanism for these phenomena.

Gene editing technology may be useful for reduction of vulnerability of becoming opioid addiction to a certain extent. However, is it reasonable for applying gene editing to prevent opioid addiction medically and ethically?

Prevention for becoming opioid addiction and treatment for patients already suffering from opioid addiction seem to require different medical procedures. Is CRISPR gene editing also useful for treatment of patients already suffering from opioid addiction?

Author Response

To Dear Reviewer3,

We appreciate your suggestions and added some content as following.

Answer to question 1:

We have added the following paragraph in the section 2. Opioid addiction, biological mechanism and medications to describe biological and molecular mechanism for desensitization, tolerance and addiction, “Volkow et al. reviewed the articles related to opioid addiction and found that the molecular mechanisms of opioid-induced tolerance and physical dependence play an important role in opioid addiction. After repeated opioid administration, the opioid receptor regulation such as desensitization and internalization started, as well as the opioid receptor tolerance, involving several cellular processes such as upregulation of cyclic AMP/protein kinase A, and cAMP response element‐binding signaling. In addition, craving for drug, diminishing self-control, and strong responses to drug-associated stimuli are associated with the cellular and molecular change of the glutamatergic projection in the prefrontal cortex and the basal ganglia region, such as alterations in G protein signaling in the prefrontal cortex and increased presynaptic glutamate release in the accumbens.”

Answer to question 2 and 3,

We have added the following paragraph in the discussion section. “If opioid addiction is an inheritable disease with genetic defective in Drd2, GHSR and Gabra2, it is reasonable for applying gene editing technologies (e.g., CRISPR) to prevent opioid addiction medically and ethically. If patients are already suffering from opioid addiction, the therapeutic outcome using CRISPR gene editing is limited unless the patients have defective genes associated with opioid addiction. The novel CRISPR/Cas9 gene editing has led to an increased interest to treat neurodegenerative diseases such as Alzheimer’s disease and Huntington’s disease. Thus, gene therapy might be effective in the treatment of opioid addiction, though most of them are currently under research and more trials/review are needed because of safety concerns and ethics issues. Regarding these cases, cell therapy based on CRISPR is recommended and potentially useful for the treatment of patients already suffering from opioid addiction.”

Again, we appreciate your review to make this manuscript more complete and more comprehensive.

Round 2

Reviewer 3 Report

The manuscript has been significantly improved to be accepted.